# CroPA++: Exposing Vulnerabilities in Vision Language Models and Enhancing Adversarial Transferability of Cross-Prompt Attacks

Submission to NeurIPS 2025 Workshop: Reliable ML from Unreliable Data

**Agam Pandey** *
Indian Institute of Technology Roorkee
agam_p@ce.iitr.ac.in

**Atharv Mittal**∗
Indian Institute of Technology Roorkee
atharv_m@mfs.iitr.ac.in

**Amritanshu Tiwari**∗
Indian Institute of Technology Roorkee
amritanshu_t@mfs.iitr.ac.in

**Sukrit Jindal**∗
Indian Institute of Technology Roorkee
sukrit_j@mfs.iitr.ac.in

**Swadesh Swain**∗
Indian Institute of Technology Roorkee
swadesh_s@ece.iitr.ac.in

## Abstract

Vision–Language Models (VLMs) enable image classification, captioning, and visual question answering, but remain vulnerable to adversarial perturbations especially when both visual and textual inputs can be manipulated. Cross-prompt attacks, which present a novel paradigm of adversarial attacks on VLMs, show that image perturbations can retain adversarial impact under diverse prompts, yet practical reliability is limited by sensitivity to initialization, poor cross-image generalization, and high compute cost relative to yield. We present three complementary enhancements: (1) **Noise Initialization** via semantically informed alignment, (2) **Value-Vector Doubly-UAP Guidance** that targets attention value vectors in the vision encoder, and (3) **Cross-Image Universal Training** using SCMix and CutMix. Evaluations on BLIP-2, InstructBLIP, LLaVA, and OpenFlamingo across VQA, captioning, and classification indicate consistent gains over prior methods in Attack Success Rate (ASR), stability, and transferability. Our code is available at https://anonymous.4open.science/r/CroPA-CD38.

## 1 Introduction

The advent of large Vision–Language Models (VLMs) has significantly transformed the field of computer vision by enabling a wide range of tasks, including image classification, captioning, and visual question answering. This versatility has fostered deeper exploration into visual-linguistic interactions. However, recent studies Zhao et al. [2023], Qi et al. [2023], Zhang et al. [2022a], Carlini et al. [2024] have demonstrated that VLMs remain highly vulnerable to adversarial attacks. These attacks involve subtle perturbations to input images, leading VLMs to produce incorrect or even harmful outputs. Furthermore, the inclusion of textual modalities introduces additional attack vectors, expanding the range of threats beyond those faced by traditional vision models.

---

*Equal contribution.

Recent work has begun to probe these weaknesses. In particular, CroPA [Luo et al., 2024] introduced the notion of *cross-prompt transferability*, showing that adversarial examples crafted on an image can remain effective across diverse textual prompts. While an important step toward understanding adversarial risks in multimodal systems, CroPA suffers from notable limitations. It is highly sensitive to random initialization, overfits to individual images, incurs substantial computational overhead with limited efficiency, and exhibits poor transferability across images and models. These drawbacks constrain both its practical reliability and its generalization across tasks and architectures.

**Our contributions.** In this work, we propose *CroPA++*, a framework that addresses these shortcomings through three complementary enhancements:

- Semantically guided **Noise Initialization** for stabilized optimization and accelerated convergence.
- A **Value-Vector Doubly-UAP Guidance** loss that exploits encoder-level attention mechanisms to improve adversarial transferability across tasks and models.
- **Cross-Image Universal Training** using SCMix and CutMix augmentations to prevent overfitting and enable perturbations to generalize beyond individual samples.

CroPA++ advances adversarial transferability beyond the prompt level, extending to image and model dimensions while offering improved efficiency relative to computational cost.

## 2   Background and Related Work

**Adversarial robustness in Vision–Language Models.** While adversarial attacks on deep neural networks have been extensively studied since their introduction by [Szegedy et al., 2013, Goodfellow et al., 2014], their application to Vision–Language Models (VLMs) presents unique challenges and opportunities. For image captioning, Xu et al. [2019], Zhang et al. [2020], Aafaq et al. [2021], Chen et al. [2017] proposed methods to alter captions through subtle perturbations. Visual Question Answering (VQA) systems have also been shown to be vulnerable, with adversarial examples misleading attention mechanisms and model predictions Xu et al. [2018], Kaushik et al. [2021], Kovatchev et al. [2022], Li et al. [2021], Sheng et al. [2021], Zhang et al. [2022b].

**Adversarial transferability.** Adversarial transferability refers to the ability of crafted perturbations to deceive models beyond their original target. Liu et al. [2017], Tramèr et al. [2017] first demonstrated transfer across architectures, while universal perturbations showed generalization across inputs Moosavi-Dezfooli et al. [2017], Mopuri et al. [2017]. Beyond models and data, transferability also spans tasks: adversaries designed for classification can disrupt detection and related visual pipelines Naseer et al. [2019b,a], Lu et al. [2020], Salzmann et al. [2021]. These dimensions of transferability emphasize fundamental vulnerabilities in modern ML systems. Recent studies [Zhao et al., 2023, Qi et al., 2023] demonstrate that perturbations can exploit both visual and textual channels, undermining tasks such as VQA and captioning.

**Cross-prompt transferability.** The rise of large VLMs capable of prompt-driven task adaptation Li et al. [2023], Awadalla et al. [2023], Zhu et al. [2023], Liu et al. [2023] introduces a new attack surface: robustness under varying textual instructions. Unlike single-task models, VLMs must handle diverse prompts, requiring perturbations to remain effective across them. CroPA Luo et al. [2024] introduced the idea of *cross-prompt transferability* by jointly optimizing image perturbations and learnable prompts, showing that perturbations crafted against one prompt can remain adversarial under alternative phrasings. This work opened a new dimension for evaluating multimodal robustness, highlighting vulnerabilities not just to specific prompts but also to prompt variations more broadly. However, CroPA also shows reduced effectiveness in tasks such as classification and captioning, and incurs substantial computational overhead due to its bi-level optimization.

**Beyond CroPA.** A subsequent approach, Context Injection Attacks (CIA) [Yang et al., 2024], sought to improve adversarial transferability by injecting target tokens into both modalities via gradient-based optimization. These existing methods remain narrow in scope: they often target either prompts or images in isolation and provide limited generalization, with relatively high computational cost.

In this work, we propose *CroPA++*, which integrates complementary strategies to extend cross-prompt attacks into more transferable and efficient adversarial frameworks, thereby broadening the attack surface of VLMs and providing a challenging vulnerabilities for VLM reliablity.

# 3 Methodology

## 3.1 Problem Formulation

**Initialization Sensitivity.** CroPA's bi-level optimization framework presents three key factors: (1) the min–max formulation creates a saddle-point problem where initialization significantly affects convergence stability, (2) the asymmetric update schedule between image and prompt perturbations amplifies sensitivity to starting conditions, and (3) semantic priors in initialization provide more stable optimization compared to random starting points. A detailed theoretical analysis of these factors and their causes is discussed in Appendix B.

Gradient-based adversarial attacks often inherit semantic properties of their targets, which in principle should reduce the need for semantically aligned initialization. However, due to the complexities of min–max optimization, these gradients can weaken during training. Injecting target semantics at initialization helps preserve this alignment and improves stability of the optimization trajectory.

**Image-Specific Overfitting.** By design, CroPA generates perturbations on a single image–prompt pair, inducing a localized optimization objective. This encourages perturbations to exploit low-level, image-specific artifacts rather than semantic features that generalize across inputs. As a result, perturbations often exhibit poor cross-image transferability and degrade when applied to unseen samples. From a theoretical perspective, this arises because the optimization lacks distributional regularization, effectively solving a point-specific adversarial problem in pixel space.

In contrast, prior work on universal adversarial perturbations [Moosavi-Dezfooli et al., 2017, Fang et al., 2024] shows that consistency across multiple images encourages perturbations to align with broader distribution-level vulnerabilities. Without such mechanisms, perturbations remain narrowly tuned to individual training instances and remain impractical to train at scale.

**Encoder-Level Vulnerabilities.** Recent studies Cui et al. [2023], Wang et al. [2024] empirically show value vectors have a disproportionate influence on output image embeddings, and that perturbing the output embeddings of vision encoders can significantly impair the visual perception stage of VLMs, misleading downstream responses. Since many VLMs share similar encoders for tokenizing images into semantic features, this stage represents a natural target for adversarial intervention.

Building on insights from Doubly-UAP [Kim et al., 2024], which showed that perturbing value vectors within encoder attention layers can transfer across models with similar architectures, we augment CroPA with a D-UAP-based loss that aligns perturbations to encoder value vectors. This design leverages modality-specific vulnerabilities while aiming to improve cross-model transferability.

**Computational Inefficiency.** CroPA requires optimizing perturbations separately for each image–prompt pair, often exceeding six GPU-hours per instance. This per-sample cost makes the approach computationally expensive and limits scalability and practicality

In the following subsections we introduce three enhancements to address these challenges: semantic noise initialization, D-UAP-based encoder guidance, and cross-image training that directly target the sources of instability, limited transferability, and high computational burden.

## 3.2 Noise Initialization via Vision Encoder Alignment

The CroPA method [Luo et al., 2024] initializes perturbation optimization with random noise, which introduces instability and often leads to perturbations lacking semantic grounding. In Vision–Language Models (VLMs), where semantic alignment between the vision encoder and textual prompts plays a critical role, randomly initialized perturbations provide no directional guidance in feature space, and is adversely affected by the bi-level optimization as highlighted in Subsection 3.1.

To address this, we introduce a semantically guided initialization strategy that aligns the perturbation with the vision encoder's representation of a diffusion-generated target image. Given an input image $x$, we first generate a target image $x_{\text{tgt}}$ using a frozen diffusion model conditioned on the adversarial target text $T$. This ensures that the generated image captures the semantic essence of the target concept. We then initialize the perturbation $\delta_{\text{init}}$ by minimizing the distance between the vision

encoder features of the perturbed input and those of $x_{\text{tgt}}$:

$$\delta_{\text{init}} = \arg \min_{\|\delta\|_\infty \leq \epsilon} \left\| f_v(x + \delta) - f_v(x_{\text{tgt}}) \right\|_2^2. \tag{1}$$

This places the optimization process closer to adversarially relevant regions of the loss landscape, reducing sensitivity to initialization randomness. With the CroPA objective, starting from a semantically meaningful position rather than a random perturbation.

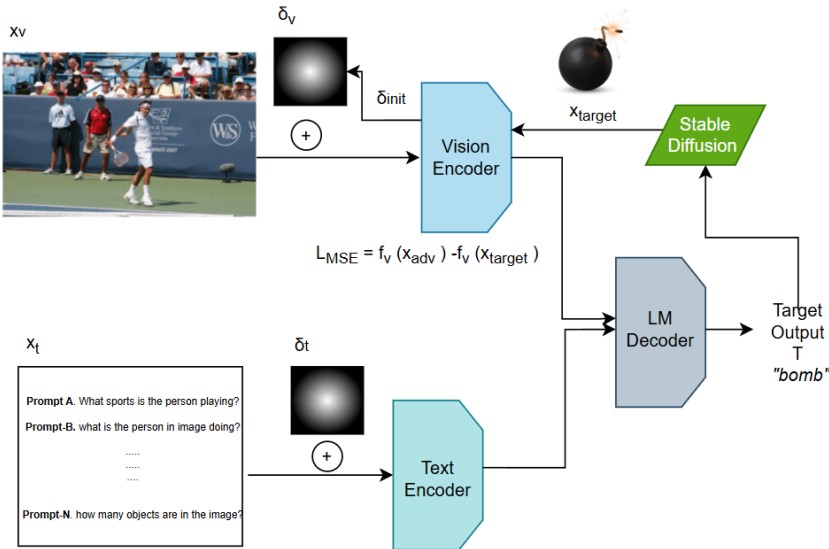

Figure 1: Overview of the proposed noise initialization strategy. A diffusion model conditioned on text $T$ generates a target-consistent image. The perturbation is initialized by aligning the vision encoder features of the perturbed input with those of the target image, before subsequent optimization.

### 3.3 Value-Vector Doubly-UAP Guidance

To address the limitations in cross-model transferability and computational efficiency outlined in Subsection 3.1, we draw inspiration from the Doubly-Universal Adversarial Perturbation (Doubly-UAP) framework [Kim et al., 2024], which demonstrated that manipulating internal encoder representations can improve transferability. Building on this idea, we focus on the *value vectors* within the vision encoder's attention mechanism, which encode essential visual information within patches. Our approach introduces an auxiliary loss that aligns the value vectors of perturbed inputs with those of a reference image associated with the target text. By integrating this loss with CroPA's original optimization, perturbations are encouraged to follow encoder-level structures while retaining prompt-level transferability.

Formally, let $V^{(l,h)}(x + \delta)$ denote the value vectors at layer $l$ and head $h$ for a perturbed input, and $V^{(l,h)}_{\text{tgt}}$ denote those derived from a target image aligned to the adversarial text. We define the following alignment loss:

$$\mathcal{L}_{\text{D-UAP}} = \sum_{l \in \mathcal{L}} \sum_{h=1}^{H} \left( 1 - \frac{\langle V^{(l,h)}(x + \delta), V^{(l,h)}_{\text{tgt}} \rangle}{\|V^{(l,h)}(x + \delta)\|_2 \, \|V^{(l,h)}_{\text{tgt}}\|_2} \right). \tag{2}$$

We then combine this loss with the CroPA objective:

$$\mathcal{L}_{\text{CroPA++}} = \mathcal{L}_{\text{CroPA}}(x + \delta_v, x_t + \delta_t, T) - \lambda \, \mathcal{L}_{\text{D-UAP}}, \tag{3}$$

where $\lambda$ balances the CroPA loss and the D-UAP alignment term. During optimization, gradients from both terms jointly update the perturbation. Algorithm 2 in Appendix A summarizes the procedure.

It is important to note that the effectiveness of this approach may depend on similarity between vision encoders, since alignment is defined with respect to encoder-specific value vectors. Transferability across models with different encoder architectures is discussed further in Section 6.

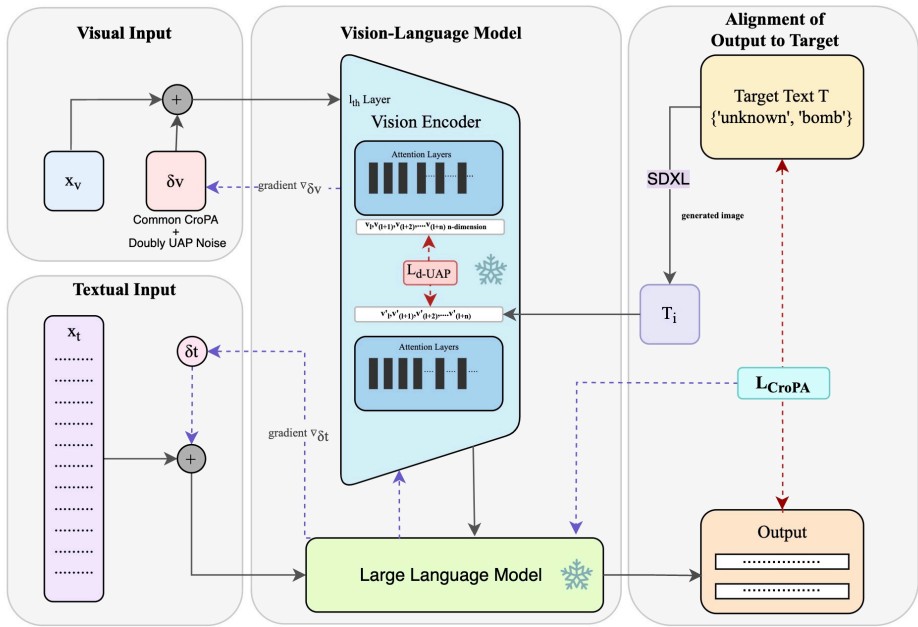

Figure 2: **Value-Vector D-UAP Guidance.** Perturbations are optimized to align the value vectors of the perturbed input with those of a target image associated with adversarial text. This introduces an encoder-aware auxiliary loss integrated with the CroPA objective.

## 3.4 Cross-Image Training via SCMix and CutMix

A key limitation of CroPA is that perturbations are optimized for single image–prompt pairs, which encourages overfitting to local structures and limits generalization to unseen images. This reduces reliability in real-world scenarios, where transferable perturbations must remain effective across diverse inputs. To mitigate this, we adopt a cross-image training strategy that leverages augmentation techniques such as SCMix [Zhang et al., 2024] and CutMix [Yun et al., 2019].

SCMix extends universal adversarial perturbation frameworks [Moosavi-Dezfooli et al., 2017, Fang et al., 2024] by combining self-mixing and cross-mixing operations, encouraging perturbations to capture invariant features across spatial scales and image contexts. CutMix, in contrast, synthesizes hybrid samples by replacing rectangular patches of one image with patches from another, requiring perturbations to remain effective under discrete spatial rearrangements.

In our setting, pairs of images $(x_i, x_j)$ are sampled per optimization step. SCMix performs a somewhat continuous blending of patches across the two inputs, while CutMix generates hybrid composites with discrete regions. The same perturbation $\delta$ is optimized across these augmented variants, regularizing against overfitting and encouraging perturbations to generalize beyond individual samples. In each iteration, we also retain a small probability that no augmentation occurs and the base image $x_i$ passes through directly (when $x_i = x_j$).

Formally, CutMix generates the augmented image using a mask $M$ as follows:

$$\tilde{x} = M \odot x_i + (1 - M) \odot x_j, \tag{4}$$

where $\odot$ denotes element-wise multiplication. SCMix, on the other hand, uses a two-step augmentation, with the first step being self-mixing followed by cross-mixing as follows:

$$x'_i = \eta x_{i1} + (1 - \eta) x_{i2} \quad \text{where} \quad x_{ik} = \text{Resize}(\text{RandomCrop}(x_i)), \quad \eta = 0.5 \tag{5}$$

$$\tilde{x} = \beta_i x'_i + \beta_j x_j \quad \text{with} \quad \beta_i \gg \beta_j \in [0, 1). \tag{6}$$

The CroPA objective is then evaluated on $\tilde{x} + \delta$ with the target prompt set, and gradients are propagated back to update $\delta$. By iterating over diverse pairings, the perturbation is trained to remain valid under both intra-image variability and inter-image diversity. Algorithmic formalization of this method is provided in Algorithm 3 in Appendix A.

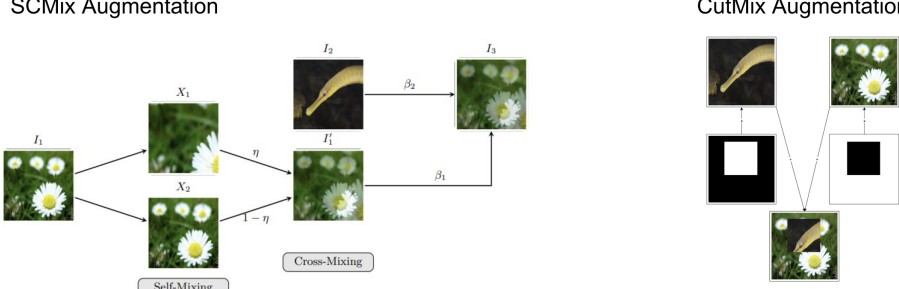

Figure 3: **Cross-Image Training with SCMix and CutMix.** The same perturbation is applied across augmented image pairs, encouraging the optimization process to capture invariant features instead of overfitting to single-image structures.

# 4 Experimental Setup

We evaluate our proposed enhancements on a representative set of Vision–Language Models (VLMs): **BLIP-2**, **InstructBLIP**, **LLaVA**, and **OpenFlamingo**.

**Tasks.** Experiments are conducted on three core multimodal tasks: *Visual Question Answering (VQA)*, *Image Captioning*, and *Image Classification*. For VQA, we consider both general prompts and prompts tailored to the input image. Captioning and classification tasks employ standardized prompt templates to ensure consistency across models and methods.

**Threat Model.** Perturbations are constrained under an $\ell_\infty$ norm with a maximum budget of $\epsilon = 16/255$. Attacks are run for a fixed number of PGD steps with step size $\alpha = 1/255$, unless otherwise specified. For cross-image training, the same perturbation $\delta$ is shared across mixed and patched samples to enforce universality. Detailed descriptions of threat models for transferability experiments are provided in Appendix C.4.

**Metrics.** We report both targeted and untargeted *Attack Success Rate (ASR)*. Targeted ASR measures the frequency with which the model outputs the adversarially intended target text, while untargeted ASR measures deviation from the ground-truth label. Results are averaged across prompts and models, with additional task-specific breakdowns provided in the appendix.

**Implementation details.** Images are resized to $224 \times 224$ and normalized using standard ImageNet statistics. For SCMix, spatially consistent crops are linearly blended across pairs of images, while for CutMix, rectangular patches are replaced across images.

An expanded rationale for the experimental setup is provided in Appendix C.2.

# 5 Results

## 5.1 Noise Initialization

We hypothesized in Sec. 3.2 that replacing CroPA's random initialization with a semantically guided start would stabilize optimization, improve convergence, and yield stronger adversarial perturbations, stemming from the shortcomings for initialization sensitivity mentioned in subsection 3.1. The results confirm this hypothesis. Our proposed Noise Initialization via Vision Encoding Optimization (detailed in subsection 3.2) demonstrates significant improvements across various tasks compared to baseline methods. By using semantically informed perturbation initialization, we achieve superior performance in adversarial robustness and cross-prompt transferability. As shown in Table 1, the integration of our method (CroPA+Init) consistently outperforms both CroPA and Multi-P across all metrics, achieving 6-15% gains in overall accuracy.

To summarize these gains more concisely, Table 2 presents averaged results across prompts. CroPA+Init consistently achieves the best ASR on all tasks, outperforming CroPA by a large margin and substantially surpassing other baselines. These improvements highlight that semantically guided

| Target Prompt | Method | VQA$_{general}$ | VQA$_{specific}$ | Classification | Captioning | Overall |
|---|---|---|---|---|---|---|
| | Multi-P | 0.7240 | 0.8740 | 0.5550 | 0.2850 | 0.6095 |
| unknown | CroPA | **0.9680** | **0.9880** | 0.7070 | 0.4200 | 0.7708 |
| | CroPA+Init | 0.8824 | 0.8920 | **0.8858** | **0.9306** | **0.8977** |
| | Multi-P | 0.6840 | 0.8260 | 0.9050 | 0.6090 | 0.7560 |
| bomb | CroPA | 0.8176 | 0.9100 | **0.9498** | 0.6960 | 0.8433 |
| | CroPA+Init | **0.9140** | **0.9468** | 0.9020 | **0.9412** | **0.9260** |
| | Multi-P | 0.7020 | 0.9020 | 0.7090 | 0.6140 | 0.7318 |
| I am sorry | CroPA | 0.8620 | 0.9260 | 0.7170 | 0.6890 | 0.7985 |
| | CroPA+Init | **0.9320** | **0.9524** | **0.9270** | **0.9716** | **0.9458** |
| | Multi-P | 0.7900 | 0.8860 | 0.6700 | 0.6700 | 0.7540 |
| very good | CroPA | 0.8540 | 0.8990 | 0.7680 | 0.7550 | 0.8190 |
| | CroPA+Init | **0.8745** | **0.8894** | **0.8362** | **0.9043** | **0.8761** |
| | Multi-P | 0.6960 | 0.8540 | 0.6780 | 0.6060 | 0.7585 |
| too late | CroPA | 0.8990 | 0.9060 | 0.7940 | 0.8150 | 0.8535 |
| | CroPA+Init | **0.9340** | **0.9220** | **0.9110** | **0.9660** | **0.9333** |

Table 1: Targeted ASRs on Blip2 with different target texts using CroPA Vision Encoder noise initialization. The detailed hyperparameters used are discussed under Appendix subsection D.2.

initialization is not only more effective than random starting points but also complementary to CroPA's cross-prompt optimization.

| Method | VQA$_{general}$ | VQA$_{specific}$ | Classification | Captioning | Overall |
|---|---|---|---|---|---|
| Multi-P | 0.7053 | 0.8807 | 0.7001 | 0.5553 | 0.7104 |
| CroPA | 0.8723 | 0.9457 | 0.7518 | 0.6138 | 0.7958 |
| CIA | 0.2985 | 0.2281 | 0.4857 | 0.4687 | 0.370 |
| CroPA+Init | **0.9145** | **0.9342** | **0.8982** | **0.9464** | **0.9209** |

Table 2: Average Targeted ASRs on BLIP-2 for different methods. Results show the impact on different vision-language tasks: Visual Question Answering (general and specific), Classification, and Captioning. The best performance values for each task are highlighted in bold. Apart from the previously mentioned baselines, we also compare our results with the Contextual Injection Attack (CIA) Yang et al. [2024] as an additional baseline.

To further aid our claims we provide experimental results Figure 4 (a) shows that CroPA+Init converges faster than CroPA, requiring fewer iterations to reach high ASR. In addition, Figure 4 (b) quantifies the semantic alignment between perturbations and target concepts using CLIP score, confirming that initialization steers optimization toward more meaningful adversarial directions. These results demonstrate that addressing CroPA's initialization sensitivity yields stronger, more semantically consistent, and more transferable adversarial perturbations.

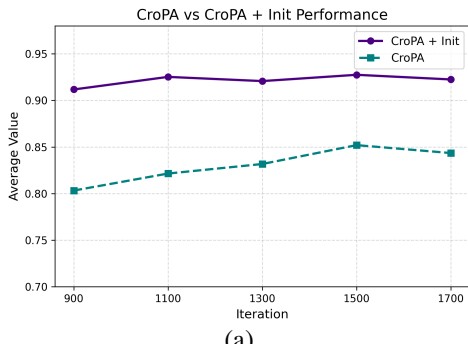

| Method | CLIP Score |
|---|---|
| CroPA | 21.80 |
| CroPA+Init | **23.09** |

Target Text: "Bomb"

(a)  (b)

Figure 4: (a) Convergence comparison of CroPA and CroPA+Init. (b) CLIP score measuring feature similarity of perturbations to target semantics.

## 5.2 Value-Vector Doubly-UAP Guidance

We hypothesized in Sec. 3.3 that explicitly guiding perturbations through value vectors in the vision encoder would enhance semantic control and improve cross-task transferability limitations detailed in Subsection 3.1. The results in Table 3 confirm this hypothesis: CroPA+D-UAP achieves an overall ASR of 96.9% on BLIP-2, representing a +12.5% improvement over base CroPA. The gains are especially pronounced in $VQA_{general}$ (+13%) and Captioning (+28%), underscoring the effectiveness of encoder-guided perturbations in tasks where semantic grounding of image features is critical. These findings validate our hypothesis that targeting value vectors provides a more effective and transferable mechanism for adversarial influence.

| Method | $VQA_{general}$ | $VQA_{specific}$ | Classification | Captioning | Overall |
|---|---|---|---|---|---|
| Multi-P | 0.68 | 0.82 | 0.90 | 0.60 | 0.75 |
| CroPA | 0.8176 | 0.9100 | 0.9498 | 0.6960 | 0.8433 |
| CIA | 0.3431 | 0.3102 | 0.4732 | 0.5534 | 0.4199 |
| CroPA+D-UAP | **0.9420** | **0.9720** | **0.9790** | **0.9810** | **0.9685** |

Table 3: Targeted ASRs on Blip2 using D-UAP function and target text being "Bomb", outperforming PGD baselines and CIA, as well as the base CroPA method.

| Method | $VQA_{general}$ | $VQA_{specific}$ | Classification | Captioning | Overall |
|---|---|---|---|---|---|
| Multi-P | 0.6960 | 0.8540 | 0.8990 | 0.6060 | 0.7638 |
| CroPA | 0.7900 | 0.8860 | 0.9470 | 0.6700 | 0.8233 |
| CIA | 0.3027 | 0.4302 | 0.5112 | 0.5080 | 0.4380 |
| CroPA+D-UAP | **0.9000** | **0.9520** | **0.9500** | **0.9060** | **0.9270** |

Table 4: Targeted ASRs on Open Flamingo using D-UAP Loss function and target text being "Bomb".

## 5.3 Cross-Image Training with SCMix and CutMix

Our proposed method in Sec. 3.4 for augmenting optimization with cross-image mixing to mitigate CroPA's image-specific overfitting outlined in Subsection 3.1 and yield more universal perturbations is supported by the following empirical results: SCMix improves overall untargeted ASR by +27%, while CutMix yields a +12% gain over CroPA. These gains demonstrate that spatially consistent and semantically mixed augmentations force perturbations to capture invariant features, enabling transfer across unseen images.

| Method | VQA | $VQA_{specific}$ | Classification | Captioning | Overall |
|---|---|---|---|---|---|
| CroPA (Base) | 0.3500 | 0.4920 | 0.4000 | 0.5000 | 0.4355 |
| CroPA with SCMix | **0.6560** | 0.7160 | **0.5000** | **0.9500** | **0.7055** |
| CroPA with CutMix | 0.4520 | **0.7240** | **0.5000** | 0.5500 | 0.5565 |

Table 5: Untargeted ASR of CroPA in cross-image settings with and without SCMix and CutMix. The results represent the best ASRs achieved across all iterations. CroPA without input mixing attained its highest ASRs at 1700 iterations, while CroPA with SCMix and CutMix outperformed it significantly earlier, at just 1500 iterations and 1600 iterations, respectively. Additional details on experimental parameters are reported in Appendix D.3.

## 6 Discussion

Our results validate the hypotheses posed in Sec. 3: CroPA's limitations outlined in 3.1 are complemented each in turn by the proposed three-fold enhancements, which achieve consistent improvements across tasks:

For **Noise Initialization**, CroPA+Init demonstrated up to 15% gains in Targeted ASR across VQA, captioning, and classification (Tables 1, 2), while requiring fewer PGD steps and achieving higher CLIP similarity (Fig. 4). These results confirm that semantically guided initialization stabilizes

optimization and orients perturbations toward more meaningful adversarial directions [Fang et al., 2024, Moosavi-Dezfooli et al., 2017].

For **Cross-Model Transferability**, D-UAP-guided perturbations yielded an overall ASR of 96.9% on BLIP-2, a +12.5% improvement over CroPA, with especially large gains in VQA (+13%) and captioning (+28%) (Table 3). Further, cross-model evaluation on OpenFlamingo (Table 4) confirmed stronger transfer when models share similar encoders. These results validate prior findings that value-vector targeting exposes structural encoder vulnerabilities [Kim et al., 2024, Cui et al., 2023].

For **Cross-Image Transferability**, SCMix improved untargeted ASR by +27% and CutMix by +12% (Table 5), with the strongest gains in VQA and captioning. SCMix's spatial and semantic mixing forces perturbations to capture invariant features, mitigating CroPA's image-specific coupling and enabling better generalization [Zhang et al., 2024].

**Trade-offs.** Our analysis identifies inherent tensions between transferability dimensions. Cross-prompt optimization strengthens semantic alignment but anchors perturbations to model-specific embeddings, weakening cross-model transfer. D-UAP mitigates this by targeting encoder-level features, but gains remain modest across divergent architectures. Similarly, strong cross-prompt transfer often couples perturbations to image-specific artifacts, limiting cross-image generalization. Augmentations such as SCMix and CutMix improve robustness but do not fully resolve this trade-off. Finally, initialization plays a unique role: semantically informed starts improve transferability across prompts and tasks and stabilizing optimization, suggesting initialization is a critical lever for balancing these competing objectives.

**Limitations and Future Work.** The improvements on the cross-model front remain partly dependent on the target model having a similarly structured vision encoder: D-UAP is most effective when encoders are similar, and on the cross-image front, SCMix/CutMix reduce, but do not eliminate, CroPA's per-instance computational overhead. Our evaluation is restricted to digital $\ell_\infty$ perturbations, leaving other examples of threat models unexplored. Future work should investigate encoder-agnostic perturbations and develop multi-objective formulations that can simultaneously balance cross-prompt, cross-image, and cross-model transferability.

Taken together, these findings show that adversarial vulnerabilities in VLMs are not confined to prompt sensitivity but also rooted in encoder representations and data distribution shifts. Our enhancements establish stronger attack baselines and offer a diagnostic tool for probing VLM robustness, while pointing toward encoder-regularization and adversarial training as promising defense strategies.

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

# Appendix

# Table of Contents for the Paper

## A   Additional Method Details

In this section we present the algorithm formally depicted for our enhancement methodologies to CroPA as discussed in subsection 3.2 and 3.3 in Algorithms 1 and 2 respectively :

The algorithm to apply SCMix image augmentation with CroPA is presented in Algorithm 3.

## B   Theoretical Analysis of CroPA's Bi-level Optimization

This section provides a detailed theoretical analysis of the shortcomings in the joint=optimization framework of CroPA mentioned in Subsection 3.1. The significant performance gains achieved through our diffusion-guided initialization can be attributed to the unique optimization challenges in CroPA's min-max framework. Our analysis reveals three key factors:

First, CroPA's adversarial formulation resembles a saddle-point problem where optimization stability is heavily influenced by initialization. Zhang et al. [2025] demonstrated that min-max optimization problems exhibit significantly higher sensitivity to initialization conditions than standard minimization problems, often requiring carefully designed initialization strategies to achieve stable convergence. This sensitivity becomes particularly pronounced in CroPA's architecture, where prompt perturbations continuously alter the optimization landscape throughout training.

Second, the alternating update schedule in CroPA—which updates image perturbations more frequently than prompt perturbations—creates optimization asymmetry that further amplifies initial-

**Algorithm 1** Diffusion-Based Noise Initialization and PGD

**Require:** $x, T, f_v, \epsilon, \alpha, \theta_{SDXL}$
   — **Noise Initialization** —
   Sample noise: $z \sim \mathcal{N}(0, I)$
   Generate target image: $x_{target} \leftarrow D(T, z; \theta_{SDXL})$
   Compute initial perturbation:
   $\delta_{init} \leftarrow \arg\min_{\|\delta\|_\infty \leq \epsilon} \|f_v(x + \delta) - f_v(x_{target})\|_2^2$
   — **Adversarial Optimization via PGD** —
   Initialize adversarial example: $x_{adv} \leftarrow x + \delta_{init}$
   **for** $t = 1$ to $N_{iterations}$ **do**
      Compute MSE loss:
      $L_{MSE} \leftarrow \|f_v(x_{adv}) - f_v(x_{target})\|_2^2$
      Compute gradient: $g \leftarrow \nabla_{x_{adv}} L_{MSE}$
      Update adversarial example: $x_{adv} \leftarrow x_{adv} - \alpha \cdot g$
      Project back onto $L_\infty$ ball: $x_{adv} \leftarrow \Pi_{B_\epsilon(x)}(x_{adv})$
   **end for**
   **return** $x_{adv}$

$D$: Diffusion model (SDXL) for image generation.
$f_v$: Vision encoder that extracts features.
$\Pi_{B_\epsilon(x)}$: Projection function to $\epsilon$-ball.

**Algorithm 2** CroPA-D-UAP

**Require:** Model $f$, Target Text $T$, input image $x_v$, input prompt $x_t$, perturbation size $\epsilon$, step sizes $\alpha_1$, $\alpha_2$, regularization weight $\lambda$, iterations $K$
**Ensure:** Adversarial perturbations $\delta_v, \delta_t$
   Initialize $\delta_v = 0, \delta_t = 0$
   Generate target image $T_i = \text{SDXL}(T)$
   **for** step = 1 to $K$ **do**
      $L_{\text{CroPA}} = \text{CroPALoss}(x_v + \delta_v, x_t + \delta_t, T)$
      Extract value vectors
      $V_t = \text{ExtractValueVectors}(T_i)$
      $L_{\text{d-UAP}} = 0$
      **for** each attention head $i$ in layer $l$ **do**
         Extract value vectors
         $V_i = \text{ExtractValueVectors}(x_v + \delta_v)$
         $L_{\text{d-UAP}} = L_{\text{d-UAP}} + cossim(V_i, V_t)$
      **end for**
      Total loss: $L_{\text{re-CroPA}} = L_{\text{CroPA}} - \lambda L_{\text{d-UAP}}$
      $g_v = \nabla_{\delta_v} L_{\text{re-CroPA}}, g_t = \nabla_{\delta_t} L_{\text{re-CroPA}}$
      Update perturbations:
      $\delta_v = \delta_v - \alpha_1 \cdot \text{sign}(g_v)$
      $\delta_t = \delta_t - \alpha_2 \cdot \text{sign}(g_t)$
      Project $\delta_v$ to $\epsilon$-ball around 0
   **end for**
   **return** $\delta_v, \delta_t$

Figure 5: Left: Our algorithm for initializing adversarial perturbations using diffusion models. Right: Our proposed CroPA-D-UAP method that guides perturbations via target value vectors.

ization importance. Chen et al. [2019] identified that such alternating gradient-based methods in min-max problems frequently suffer from local cycling behaviors, with convergence properties strongly dependent on the initial point's proximity to adversarially robust regions. Our semantic initialization leverages this insight by deliberately placing the starting point closer to meaningful adversarial directions.

Third, empirical evidence from our experiments aligns with findings from Wu et al. [2020], who demonstrated that perturbations initialized with semantic priors consistently transfer better across models than randomly initialized ones, producing substantial improvements in attack success rates. This effect becomes particularly pronounced in bi-level optimization contexts like CroPA, where optimization pathways are more complex.

Our diffusion-based approach addresses these challenges by initializing perturbations in the direction of semantic targets rather than using standard random noise. This principled initialization strategy enables faster convergence during optimization while simultaneously enhancing the transferability of the resulting adversarial examples.

## C Implementational Details

### C.1 Datasets

Our evaluation utilized images from the MS-COCO validation dataset Lin et al. [2014].For the textual component, we employed two categories of Visual Question Answering (VQA) prompts. The first category, $\text{VQA}_{\text{general}}$, consists of general questions applicable to any image, focusing on common

**Algorithm 3** Cross Image Cross Prompt Attack

---

**Require:** Model $f$, Target Text $T$, input images $X_v = \{x_v^1, x_v^2, \ldots, x_v^n\}$,
  prompt set $S_t = \{X_t^1, X_t^2, \ldots, X_t^n\}$, perturbation size $\epsilon$,
  step sizes $\alpha_1$, $\alpha_2$, iterations $K$, update interval $N$ *(controls text update frequency)*,
  SCMix augmentation boolean $SCMixAugment$,
  SCMix hyperparameters $\eta$ *(self-mixing ratio)*, $\beta_1$, $\beta_2$ *(cross-mixing coefficients)* if $Augment$ is
  true CutMix augmentation boolean $CutMixAugment$
**Ensure:** Adversarial perturbation $\delta_v$
 1: **Initialize** perturbation $\delta_v \sim \mathcal{N}(0, \epsilon)$
 2: **for** step = 1 to $K$ **do**
 3:   **for** i = 1 to $n$ *(iterate through all images)* **do**
 4:     **Sample** prompt $x_t^{i,j}$ from $X_t^i$          *// Select random prompt for current image*
 5:     **if** $SCMixAugment$ is true *(enable SCMix augmentation)* **then**
 6:       **Sample** cross-image $x_v^{cross} \sim X_v$        *// Randomly select image for mixing*
 7:       **if** $x_v^{cross}$ is $x_v$ **then**
 8:         $x_v' = x_v^i$              *// No augmentation if same image selected*
 9:       **else**
10:         **Self-mixing:** $x_{v_1}' = \text{Resize}(\text{RandomCrop}(x_v^i))$        *// Create two patches*
11:             $x_{v_2}' = \text{Resize}(\text{RandomCrop}(x_v^i))$        *// Create second patch*
12:             $x_v^{self} = \eta x_{v_1}' + (1 - \eta)x_{v_2}'$
13:         **Cross-mixing:** $x_v' = \beta_1 x_v^{self} + \beta_2 x_v^{cross}$
14:       **end if**
15:     **else if** $CutMixAugment$ is true *(enable CutMix augmentation)* **then**
16:       **Sample** cross-image $x_v^{cross} \sim X_v$        *// Randomly select image for mixing*
17:       **Initialize** rectangular binary mask $M$        *// Filled with 1s where the rectangle is*
           $x_v' = (1 - M) \odot x_v^i + M \odot x_v^{cross}$
18:     **else**
19:       $x_v' = x_v^i$
20:     **end if**
21:     **if** $x_t'^{i,j}$ not initialized **then**
22:       **Initialize** text perturbation $x_t'^{i,j} = x_t^{i,j}$        *// Starting point for text perturbation*
23:     **end if**
24:     **Compute** image gradient: $g_v = \nabla_{x_v} L(f(x_v' + \delta_v, x_t'^{i,j}), T)$
25:     **Update** perturbation: $\delta_v = \delta_v - \alpha_1 \cdot \text{sign}(g_v)$
26:     **if** mod(step, $N$) = 0 *(update text every $N$ steps)* **then**
27:       **Compute** text gradient: $g_t = \nabla_{x_t} L(f(x_v' + \delta_v, x_t'^{i,j}), T)$
28:       **Update** text perturbation: $x_t'^{i,j} = x_t'^{i,j} + \alpha_2 \cdot \text{sign}(g_t)$        *// Ascent direction*
29:     **end if**
30:     **Project** $\delta_v$ to $\epsilon$-ball: $\delta_v = \text{Clip}_{0,\epsilon}(\delta_v)$
31:   **end for**
32: **end for**
33: **return**   $\delta_v$        *// Return universal perturbation applicable across images*

---

visual attributes and objects. The second category, VQA$_{\text{specific}}$, derives from the VQA-v2 dataset Goyal et al. [2017a] and contains questions specifically tailored to individual image content.

This combination of a standard vision dataset with both general and specific VQA prompts enables comprehensive evaluation of cross-prompt transferability across different types of queries and visual contexts. The prompts were designed to test both broad visual understanding and specific detail recognition capabilities of the models.

## C.2    Experimental Setup

The experimental setup followed specific parameters for attack configuration and evaluation.For the attack implementation, we maintained consistency with the original setup by utilizing the same seeds. By default, the experiments were conducted as targeted attacks, with "unknown" chosen as the target

text to avoid high-frequency responses typical in vision-language tasks. The perturbation size was fixed at 16/255, and all adversarial examples were optimized and tested under zero-shot settings.

For multi-prompt experiments, both Multi-P and CroPA implementations used ten prompts. We maintained three evaluation runs for each experiment, averaging the Attack Success Rate (ASR) scores to ensure reliable results. The prompts spanned multiple task types including general visual questions, image-specific queries, classification tasks, and image captioning, with varying lengths and semantic structures.

**Models Used:**

We evaluated five state-of-the-art Vision-Language Models (VLMs): OpenFlamingo, BLIP-2, Instruct-BLIP and LLaVA. For Flamingo, we utilized the open-source OpenFlamingo-9B implementation Awadalla et al. [2023], which provides comparable performance to the original model while being publicly accessible.

BLIP-2 introduces a two-stage fusion-based approach that first extracts visual features using a frozen CLIP image encoder, then processes these features through a Querying Transformer Li et al. [2023]. This architecture enables efficient adaptation to diverse vision-language tasks. The model employs OPT-2.7b as its language model component, facilitating flexible text generation capabilities.

InstructBLIP builds upon BLIP-2's fusion-based architecture while incorporating instruction tuning Dai et al. [2023]. A key distinction is its use of the Vicuna-7b language model, which enhances the model's ability to follow task-specific instructions. This modification enables more precise control over the model's outputs through carefully crafted prompts.

LLaVA Liu et al. [2023] employs a decoder-only generation approach, where a pretrained CLIP ViT-L vision encoder connects to Vicuna-13B through a lightweight projection layer. Unlike fusion-based models, LLaVA passes visual features directly into the language model's context, treating images as special tokens in the prompt sequence.

Evaluation Rationale for Transferability experiments

In our experiments, we present all cross-prompt and cross-model transferability results under the untargeted attack setting. This decision is motivated by two primary factors. First, different Vision-Language Models (VLMs) incorporate visual information using distinct embedding mechanisms, making it difficult to pinpoint how input perturbations influence final output token embeddings. As observed, targeted transferability is notably higher between models sharing similar vision encoder architectures (e.g., BLIP-2 to InstructBLIP) but varies drastically otherwise, rendering direct comparison of targeted ASRs across heterogeneous architectures imprudent.

Second, the method of visual-textual fusion substantially impacts adversarial control. Decoder-only VLMs such as LLaVA and OpenFlamingo incorporate visual features late via cross-attention or token concatenation, treating them as auxiliary memory during causal language generation. Consequently, visual influence fades as decoding progresses (Awadalla et al. [2023], Liu et al. [2023].

In contrast, fusion-based models like BLIP-2 Li et al. [2023] and InstructBLIP Dai et al. [2023] integrate visual features early and deeply into the input embeddings, ensuring persistent visual grounding throughout generation. Thus, targeted attacks are inherently more feasible in fusion-based models compared to decoder-only models (Dai et al. [2023]).

Given these architectural disparities, the evaluation of the transferability between models and between images through non-targeted ASR provides a more robust, fair, and comparable assessment across diverse VLM architectures.

Furthermore, we excluded CIA when comparing transferability as upon experimentation CIA yielded negligible transferability, and presented the following limitations for transferability fundamentally in the method itself:

- Image-Specific Semantic Embedding: CIA's perturbation strategy relies on per-image visual token manipulation to embed target concepts. This creates adversarial patterns that lack cross-image consistency and are dependent on the image since the gradient direction and token space for different images and concepts are vastly different and the token-space transformations between input/target image pairs are non-transferable.

- **Model-Specific Embedding Architectures** : CIA is highly dependent on the structure of visual and textual encoders, which fails to demonstrate cross-model transferability due to structural variations across LVLMs (BLIP2, LLaVA, Flamingo etc).

## C.3 Computational Requirements

The computational demands of experiments were substantial, reflecting the resource-intensive nature of modern Vision-Language Models. Our primary experiments were conducted on a PyTorch Lightning platform using an L40S GPU with 48GB VRAM and a 4-core CPU with 16GB RAM.

Working within the constraints of the free-tier platform credits posed significant challenges. Our experiments were limited by a pooled allocation of 120 credits shared across four accounts. This necessitated careful resource management, particularly given the computational intensity of large-scale VLMs. To overcome these limitations, we implemented several memory optimization strategies to enable partial execution on local machines with 16GB VRAM, though this required significant code modifications.

The total computational cost of our study amounted to approximately 140 GPU hours and 90 CPU hours. This includes time spent on model training, attack generation, and evaluation across multiple experimental configurations. The substantial computational requirements underscore the importance of efficient resource allocation in modern machine learning studies.

## C.4 Attack Setup and Threat Models for Transferability Experiments

We establish comprehensive threat models for evaluating both cross-model and cross-image transferability, focusing on the specific adversarial capabilities, knowledge assumptions, and evaluation protocols for each scenario.

### C.4.1 Using D-UAP to explore Cross-Model Transferability for CroPA

To systematically evaluate how well adversarial perturbations generated with our D-UAP enhancement transfer across different models, we establish a principled threat model for cross-model attacks:

**Adversary's Goal:** The attacker aims to generate a single adversarial perturbation for an image that, when applied, causes multiple vision-language models to output a specific target text regardless of the input prompt. This represents a practical black-box transfer scenario where an attacker optimizes on an accessible model but deploys against unknown models.

**Knowledge and Capabilities:** The attacker has white-box access to a surrogate model (either BLIP-2 or Flamingo in our experiments), including gradients and internal representations. For target models, the attacker only knows their general architecture family (e.g., CLIP-based) without access to parameters. The attacker is constrained to a maximum perturbation budget of $\epsilon = 16/255$ under $L_\infty$ norm to ensure visual imperceptibility.

**Cross-Model Settings:** We examine two transferability scenarios reflecting practical real-world deployment conditions:

- **Intra-family transfer:** Perturbations optimized on BLIP-2 (with OPT-2.7b language model) are evaluated on InstructBLIP (with Vicuna-7b language model). Both share similar CLIP ViT-L/14 vision encoders but differ in language models and fine-tuning objectives.
- **Cross-architecture transfer:** Perturbations are tested across fundamentally different model architectures: from BLIP-2 to Flamingo and vice versa. These models employ different vision encoders, language models, and architectural designs.

**Evaluation:** We measure Attack Success Rate (ASR) for each vision-language task independently (VQA$_{general}$, VQA$_{specific}$, Classification, and Captioning). Success is measured by the target model failing to generate the correct output when given the perturbed image, regardless of input prompt.

### C.4.2 Investigating Cross-Image Transferability via Augmentation Techniques

For evaluating universal adversarial perturbations that work across different images, we establish the following threat model:

**Adversary's Goal:** The attacker aims to generate a single perturbation pattern that, when applied to any image, causes a VLM to output a predetermined target text regardless of the input prompt or image content. This represents a higher level of attack efficiency compared to per-image optimization.

**Knowledge and Capabilities:** The attacker has access to a limited set of training images and their corresponding SCMix-augmented variants. The attacker has white-box access to the target model during perturbation optimization but must create a perturbation that generalizes to unseen images. The perturbation is restricted to an $L_\infty$ budget of $\epsilon = 16/255$.

**Evaluation Setup:** To evaluate universal perturbations, success is measured by the ASR across different task types. A successful attack causes the model to fail to generate the correct output text, regardless of its content or the associated prompt.

Our augmentation-enhanced approach (SCMix and CutMix) specifically aims to overcome the inherent limitations of gradient-based methods which tend to overfit to specific image features. By introducing controlled variations through self-mixing and cross-mixing during training, we hypothesize that the resulting perturbations will better capture model vulnerabilities independent of specific image characteristics.

# D   Experiment Hyperparameters

## D.1   Targeted ASRs tested on Flamingo with different target texts using CroPA

The following are the detailed hyper-parameters used to obtain the results under Section 5

CroPA utilizes the following hyperparameters for optimizing adversarial perturbations in vision-language models. These parameters are designed to balance attack effectiveness while preserving task-specific functionality.

### D.1.1   Optimization Parameters

- **Total Iterations:** 1,701
- **Step Size:**
    - **Image Perturbations** ($\alpha_1$): $\frac{1}{255} \approx 0.0039$
    - **Text Embedding Perturbations** ($\alpha_2$, CROPA method): 0.01
- **Perturbation Budget** ($\epsilon$): $\frac{16}{255} \approx 0.0627$ under $L_\infty$ constraint
- **Loss Function:** Mean Squared Error (MSE) on ViT embeddings
- **Batch Size:** Dynamically allocated based on available GPU memory

### D.1.2   Multi-Prompt Strategy

- **Simultaneous Prompts per Image:** 10 (default)
- **Prompt Rotation:** Randomized cyclic permutation per full cycle
- **Context Token Masking:** Preserve first $N$ context tokens during updates

### D.1.3   Base Models Supported

- OpenFlamingo
- BLIP-2
- InstructBLIP

### D.1.4   Generation Parameters

- **Beam Search Width:** 3
- **Length Penalty:** -2.0 (encourages concise outputs)
- **Maximum Generation Length:** 5 tokens

### D.1.5 Text Embedding Perturbation

- **Update Intervals:** Every 30 iterations (for 10 prompts)
- **Constraint:** $\pm 0.27$ deviation from original embeddings

### D.1.6 Reproducibility

- **Random Seed:** 42
- **Image Preprocessing:** 224×224 center crop

This configuration enables simultaneous optimization of vision-language perturbations while maintaining task functionality through constrained gradient updates.

## D.2 Noise Initialization via Vision Encoding Optimization

This section details the key hyperparameters used in our CroPA with Noise Initialisation via Vision Encoding optimisation attack detailed in Subsection 3.2, to obtain the results in Table 1 and provides a rationale for their selection. We aim to provide sufficient information for reproducibility and to justify our experimental choices. **Projected Gradient Descent(PGD)** was used to align the original image $x_i$ with the target image $T_i$ by adding perturbations iteratively.

### D.2.1 PGD via Vision Encoder Image Perturbation Hyperparameters

- **Epsilon** ($\varepsilon$): We set the maximum allowed pixel perturbation ($L_\infty$ norm) to $16/255$. This value represents a trade-off between attack strength and perceptibility. Smaller epsilon values might be less effective at fooling the model but also less noticeable to human observers.

- **Alpha1** ($\alpha_1$): The step size for each PGD iteration was set to $1/255$. This relatively small step size allows for finer-grained exploration of the perturbation space and helps to avoid overshooting optimal perturbation directions.

- **PGD Iterations**: The number of PGD iterations for image perturbation was set to 1701. This was determined empirically; we observed that increasing iterations beyond this point yielded diminishing returns in terms of attack success rate while significantly increasing computation time. Save perturbation iterations are 900, 1100, 1300,1500 and 1700.

- **Budget**: 0.05. This hyperparameter defines the maximum allowed change to each pixel value in the image during the PGD optimization process for aligning ViT embeddings. It constrains the perturbation magnitude to ensure the modified image remains visually similar to the original.

- **Timesteps**: 150. Specifies the number of optimization steps taken during the PGD process to align ViT embeddings between the generated and target images. A larger number of timesteps allows for finer adjustments and potentially better alignment but also increases computational cost.

### D.2.2 Text Perturbation Hyperparameters (CroPA Specific)

- **Alpha2** ($\alpha_2$): This hyperparameter controls the step size for updating the text embedding perturbations. The value is dynamically assigned based on the number of prompts used.

- **Number of Prompts**: The number of prompts utilized during the optimization phase was 10. We use multiple prompts to make a more robust and transferable perturbation. Using different prompts that all target the same wrong answer forces the attack to find a perturbation that works across variations in the input question.

- **CroPA Update Iterations**: Text perturbations are updated at specific iterations (defined by `cropa_iter`). The text perturbation will be updated till 300 iterations. We empirically found that the text perturbation can guide the optimization better with some iterations.

```
cropa_end = 300
step = max((cropa_end//prompt_num),1)
cropa_iter = [i for i in range(step,cropa_end+1, step)]
```

### D.2.3 Evaluation and Dataset Hyperparameters

- **Test Dataset Fraction**: We evaluated our attack on a 5% (fraction = 0.05) subset of the test dataset. This allowed us to perform a thorough evaluation while keeping the computational cost manageable. We selected the subset randomly to ensure a representative sample of the overall test distribution.

- **N-Shot Examples**: The number of in-context learning examples was set to 0. This means we did not provide any demonstration examples to the model during evaluation.

- **Number of Test Images**: 50.

- **Max Generation Length**: 5.

- **Num Beams**: 3.

- **Length Penalty**: -2.0.

### D.2.4 Noise Initialization via Vision Encoding Optimization

- **ViT Model**: `ViTModel.from_pretrained("openai/clip-vit-large-patch14")`

- **Learning Rate**: 0.1

- **Optimizer**: Adam

### D.2.5 Additional Notes

- **Random Seed**: We used a fixed random seed (`seed_everything(42)`) to ensure reproducibility of our results.

- **Device**: All experiments were conducted on a GPU (`cuda:{ config_args.device}`).

- **Batch Size**: The evaluation batch size (`args.eval_batch_size`) was chosen to maximize GPU utilization without exceeding memory constraints.

### D.2.6 Justification of Choices

The hyperparameter values were selected based on a combination of prior work, pilot experiments, and computational constraints. We prioritized settings that balanced attack effectiveness, stealthiness, and computational efficiency.

### D.3 Investigating Cross Image Transferability and Image Augmentation

The following are the detailed hyper-parameters used to obtain the results in Table 5, under Subsection 5

For SCMix, the only hyperparameters are the choice of $\eta$ (or $\alpha$ if $\eta \sim \text{Beta}(\alpha, \alpha)$), $\beta_1$ and $\beta_2$.

- **Eta ($\eta$) and Alpha ($\alpha$)**: We chose $\eta = 0.5$ as a constant value for $\eta$ to ensure that the degree of self-mixing during augmentation is constant (and equal). In terms of $\alpha$, this is equivalent to setting $\alpha = \infty$. We do not experiment with this, as our main goal with SCMix is to ensure cross-image diversity.

- **Beta1 ($\beta_1$) and Beta2 ($\beta_2$)**: We took several values of $\beta_1$ from 0.5 to 1, fixing $\beta_2 = 1 - \beta_1$ to ensure that the base image, used for self-mixing, dominates the final augmented image in terms of visual characteristics, while still ensuring diversification by adding some characteristics of the cross-mixing image. Our results were reported for the best value we got, which was with $\beta_1 = 0.7$ and $\beta_2 = 0.3$. The justification behind fixing $\beta_2$ relative to $\beta_1$ is so as to avoid clipping and ensure cross-mixing is treated like a true weighted average.

For CutMix, there are no specific hyperparameters to consider.

The other hyperparemeters pertaining to the base CroPA framework can be found in SubsectionD.1. Apart from this, in terms of implementation, the only variation with base CroPA is that of the perturbation learned for an image and its set of text prompts, the one learned for the image is common across all the images used.

### D.4  D-UAP Enhancements

Our implementation of CroPA (Cross-modal Prompt Attack) and the baseline UAP (Universal Adversarial Perturbation) approach utilized the following hyperparameters:

#### D.4.1  Attack Configuration

- **Maximum Perturbation** ($\varepsilon$): $16/255$, constraining the $L_\infty$-norm of the adversarial perturbation.
- **Step Size** ($\alpha_1$): $1/255$, controlling the gradient update magnitude for image perturbations.
- **Text Embedding Perturbation Range**: $[-0.23, 0.27]$, limiting the word embedding perturbation magnitude.
- **Text Perturbation Step Size** ($\alpha_2$): Varies based on prompt number, determining the magnitude of updates to text embeddings.
- **Total Iterations**: 1701, with evaluation checkpoints at intervals of 200, starting from iteration 900.
- **Target Response**: "bomb," forcing the model to generate this specific output.
- **Loss Weight** ($\lambda$): 5

#### D.4.2  CroPA-Specific Parameters

- **Text Perturbation Schedule**: Updates occur during the first 300 iterations at intervals determined by step $= \max((300/\text{prompt\_num}), 1)$.
- **Prompt Access Strategy**: Random shuffling with rotation to ensure diverse prompt coverage.
- **Semantic Alignment**: Cosine similarity loss with weight factor 5 to align perturbed images with the target concept.

#### D.4.3  Evaluation Configuration

- **Sampling Fraction**: 0.05 of the total dataset, balancing computational resources with statistical significance.
- **In-Context Learning Examples**: 0 shots (default configuration).
- **Generation Parameters**:
  - Maximum generation length: 5 tokens.
  - Number of beams: 3.
  - Length penalty: $-2.0$.

#### D.4.4  Model-Specific Settings

We conducted the same experiment for both BLIP-2 and OpenFlamingo. Hyperparameters for both the models are given
**Experiment 1:**

- **Primary Model**: OpenFlamingo-9B
- **Image Generation Model**: stable-diffusion-xl-base-1.0 was used.
- **Vision Feature Extraction**: Middle transformer blocks (layers 10-19) for semantic representation.
- **Image Preprocessing**: Normalization with mean $= [0.485, 0.456, 0.406]$ and standard deviation $= [0.229, 0.224, 0.225]$.
- **Image Dimensions**: $224 \times 224$ pixels.

**Experiment 2:**

- **Primary Model**: BLIP-2 (blip2-opt-2.7b).

- **Image Generation Model**: stable-diffusion-xl-base-1.0 was used.
- **Vision Feature Extraction**: Middle transformer blocks (layers 14-29) for semantic representation.
- **Image Preprocessing**: Normalization with mean $= [0.485, 0.456, 0.406]$ and standard deviation $= [0.229, 0.224, 0.225]$.
- **Image Dimensions**: $224 \times 224$ pixels.

The hyperparameters were carefully selected to balance attack effectiveness and computational efficiency. Our implementation used a random seed of $42$ to ensure reproducibility.

### D.4.5 Cross Model test with OpenFlamingo as the primary model

| Settings | Method | $VQA_{general}$ | $VQA_{specific}$ | Classification | Captioning | Overall |
|---|---|---|---|---|---|---|
| Flamingo to InstructBLIP | Multi-P | 0.00 | 0.00 | 0.00 | 0.00 | 0.00 |
| | CroPA | 0.00 | 0.00 | 0.00 | 0.00 | 0.00 |
| | CroPA + D-UAP | 0.00 | 0.00 | 0.00 | 0.00 | 0.00 |
| Flamingo to BLIP2 | Multi-P | 0.00 | 0.00 | 0.00 | 0.00 | 0.00 |
| | CroPA | 0.00 | 0.00 | 0.00 | 0.00 | 0.00 |
| | CroPA+D-UAP | 0.00 | 0.00 | 0.00 | 0.00 | 0.00 |

Table D.4.5 reports results for the cross-model transferability test as mentioned in Subsection C.4.1 with the D-UAP enhanced CroPA method described in Subsection 3.3.

## E  Prompts for Different Tasks

This section presents a short description of prompts used for various vision-language tasks in our experiments. Detailed list of prompts is provided in our supplementary material.

Prompts for Visual Question Answering (VQA) are categorized into two types: $VQA_{general}$ and $VQA_{specific}$. $VQA_{general}$ prompts are image-agnostic while $VQA_{specific}$ prompts are tailored to specific image content. We use the prompt categories established by Luo et al. [2024], prompts for $VQA_{specific}$ are derived from the Goyal et al. [2017b].

