# OpenReview forum: "CroPA++: Exposing Vulnerabilities in Vision Language Models and Enhancing Adversarial Transferability of Cross-Prompt Attacks"
_NeurIPS.cc/2025/Workshop/Reliable_ML — NeurIPS 2025 - Reliable ML Workshop_

### Official Review · Reviewer_UMna · 2025-09-20
**Improving CroPA**

**Rating:** 6
**Confidence:** 1

**Review:**

*Summary*
The paper proposes CroPA++, three enhancements to cross-prompt attacks on VLMs: (1) Semantically Guided Noise Initialization: start from semantics-aligned noise to reduce variance and speed convergence; (2) Value-Vector Doubly-UAP Guidance: optimize a universal perturbation by steering attention value vectors toward adversarial targets (and away from benign ones) for stronger cross-prompt consistency; and (3) Cross-Image Universal Training: learn a single perturbation over mixed/composited images to improve regularization and transfer to unseen images and prompts.

*Strengths*
 Limitations of prior CroPA (initialization sensitivity, overfitting, compute cost) are explicitly identified and each is targeted by one of the three components.

*Weaknesses and suggestions*
 The bi-level optimization used in CroPA is referenced in multiple places but not stated formally; a concise, self-contained definition would help unfamiliar readers.

While the three enhancements appear modular and can be used together, experiments are mainly reported per component; adding results for the full CroPA++ would make the cumulative benefit clearer.

Introduction can be expanded to include motivating examples, and to have a softer transition to the technical details.

---

### Official Review · Reviewer_nedm · 2025-09-20
**Review of CroPA++**

**Rating:** 6
**Confidence:** 2

**Review:**

The paper investigates adversarial attacks on VLMs. The paper builds on top of a previous method -- CroPA. The authors present several weaknesses of CroPA and propose CroPA++ with several new technical designs to address these: semantically guided noise initialization; Value-Vector Doubly-UAP Guidance loss; and universal adversarial training. As a result, CroPA++ achieves clear gains over CroPA across datasets and model backbones.

Strengths:
- The problem of VLM vulnerabilities is important and impactful
- The experiments span a variety of tasks and models
- The performance gains are quite significant

Weaknesses:
- Presentation can be improved. The problem details can be presented more in the introduction. The title of the paper seems too general and does not well reflect the contents.
- Some proposed techniques seem incremental and directly imported from previous papers. Therefore, the overall novelty is limited.
- No analysis that the proposed initialization can address the problem of initialization sensitivity.
- No sensitivity analysis
- Can the proposed techniques be combined together to yield stronger, more universal perturbations?